# The Effects of Acetazolamide on Cerebral Hemodynamics in Adult Patients with an Acute Brain Injury: A Systematic Review

**DOI:** 10.3390/brainsci13121678

**Published:** 2023-12-06

**Authors:** Claudia Stella, Anas Hachlouf, Lorenzo Calabrò, Irene Cavalli, Sophie Schuind, Elisa Gouvea Bogossian, Fabio Silvio Taccone

**Affiliations:** 1Department of Intensive Care, Hôpital Universitaire de Bruxelles (HUB), Université Libre de Bruxelles (ULB), Route de Lennik, 808, 1070 Bruxelles, Belgium; claudiastella24@gmail.com (C.S.); anas.hachlouf@ulb.be (A.H.); lorenzo.calabro94@gmail.com (L.C.); irene.cavalli@studio.unibo.it (I.C.); sophie.schuind@hubruxelles.be (S.S.); elisagobog@gmail.com (E.G.B.); 2Department of Anesthesia and Intensive Care, Policlinico Universitario Gemelli, Università Cattolica del Sacro Cuore, Largo Francesco Vito 1, 00168 Rome, Italy; 3Department of Neurosurgery, Hôpital Universitaire de Bruxelles (HUB), Université Libre de Bruxelles (ULB), Route de Lennik, 808, 1070 Bruxelles, Belgium

**Keywords:** acetazolamide, brain injury, cerebral blood flow, cerebral oxygenation, intracranial pressure

## Abstract

Background: Acetazolamide is a non-competitive inhibitor of carbonic anhydrase, an enzyme expressed in different cells of the central nervous system (CNS) and involved in the regulation of cerebral blood flow (CBF). The aim of this review was to understand the effects of acetazolamide on CBF, intracranial pressure (ICP) and brain tissue oxygenation (PbtO_2_) after an acute brain injury (ABI). Methods: Following the Preferred Reporting Items for Systematic Reviews and Meta-Analyses statement (PRISMA), we performed a comprehensive, computer-based, literature research on the PubMed platform to identify studies that have reported the effects on CBF, ICP, or PbtO_2_ of acetazolamide administered either for therapeutic or diagnostic purposes in patients with subarachnoid hemorrhage, intracerebral hemorrhage, traumatic brain injury, and hypoxic-ischemic encephalopathy. Results: From the initial search, 3430 records were identified and, through data selection, 11 of them were included for the qualitative analysis. No data on the effect of acetazolamide on ICP or PbtO_2_ were found. Cerebral vasomotor reactivity (VMR—i.e., the changing in vascular tone due to a vasoactive substance) to acetazolamide tends to change during the evolution of ABI, with the nadir occurring during the subacute stage. Moreover, VMR reduction was correlated with clinical outcome. Conclusions: This systematic review showed that the available studies on the effects of acetazolamide on brain hemodynamics in patients with ABI are scarce. Further research is required to better understand the potential role of this drug in ABI patients.

## 1. Introduction

Acetazolamide is a non-competitive inhibitor of carbonic anhydrase, the enzyme responsible for the reversible reaction of the hydration of carbon dioxide and dehydration of carbonic acid [1]. In the central nervous system (CNS), the carbonic anhydrase is expressed by different cells (i.e., neurons, oligodendrocytes, astrocytes, and choroid plexus cells), and it is involved in the regulation of intracranial pressure (ICP) through direct and indirect effects on cerebral spinal fluid (CSF) production and pH homeostasis (Figure 1) [2]. Indeed, bicarbonate reduction caused by the inhibition of carbonic anhydrase can interfere with some of the ionic channels directly involved in CSF production, which can decrease up to 48% when almost all enzymes are inhibited in the choroid plexus [1,2,3]. This is the rationale behind the off-label use of acetazolamide in conditions such as idiopathic intracranial hypertension, where the control of ICP is achieved by reducing CSF production [1]. In fact, although this direct effect has not been clearly demonstrated at therapeutic doses in humans, it has been observed in a model of healthy rats that the administration of acetazolamide reduces CSF production with a direct effect on ICP, which was eventually decreased after an initial transient increase [3]. These results are consistent with another study reporting that acetazolamide administration was associated with a decreased number of ICP spikes in rats with intracerebral hemorrhage (ICH), likely as a result of reduced CSF production and improved cerebral compliance [4]. Through very similar mechanisms, the administration of acetazolamide also leads to a reduction in the production of vitreous humor in the eye, thereby decreasing intraocular pressure [1]. The potential inhibition of aquaporin channels into the brain could also contribute to minimize cerebral edema, although these effects are mainly shown in the experimental setting [2].

Another mechanism of action of acetazolamide on the CNS is the acidification of extracellular and, possibly, intracellular milieu [5,6], which results in cerebral vasodilation. This phenomenon is secondary to the cerebral vasomotor reactivity (VMR), i.e., cerebral blood flow (CBF) variation induced by the changes in arteriolar diameter, which may happen in response to arterial CO_2_ pressure and brain tissue pH changes [7]. In both animals [6,8] and human studies [9,10], a significant increase in CBF was measured in healthy subjects within a few minutes of acetazolamide administration. Historically, various methods have been employed to measure variations in CBF, including single-photon emission computed tomography (SPECT), flow-sensitive alternating inversion-recovery (FAIR) perfusion magnetic resonance imaging (MRI), continuous arterial spin labeling (CASL) MRI, and positron emission tomography (PET) [9]. However, among these, xenon–computed tomography (Xe-CT) and transcranial Doppler (TCD) remain pertinent in both research and clinical settings. Xe-CT relies on the inhalation of a gas mixture containing oxygen and non-radioactive xenon, an extremely lipophilic inert gas that swiftly traverses the blood–brain barrier. Combined with anatomical CT images, this enables the quantitative evaluation of global and regional CBF [11]. On the other hand, TCD is based on the premise that vasomotor reactivity (VMR) primarily affects arterioles and has a negligible impact on the diameter of larger cerebral vessels, such as the middle cerebral artery. Consequently, variations in velocity detected at this level correlate with changes in CBF [7,9]. This increase in CBF and intracerebral volume could also potentially increase ICP [2].

A third effect of acetazolamide is the improvement in brain oxygenation. In six patients without brain injury, Vorstrup et al. [10] observed an increase in CBF and venous oxygen saturation, measured via a jugular bulb catheter. Similar results were obtained when acetazolamide was administered to seven patients with cerebral ischemia and regional oxygen saturation (rSO_2_, measured using near-infrared spectroscopy (NIRS) [12]. However, brain oxygenation is more accurately measured using the brain oxygen pressure (PbtO_2_) catheter in acute brain injury (ABI) patients [13], while venous oxygen saturation and rSO_2_ have several limitations in this setting [14].

In view of the important effects described above, understanding the role of acetazolamide in the management of ABI patients with different types of injury is of pivotal relevance. Thus, the aim of this study was to perform a systematic review of the literature to report the effects of acetazolamide on CBF, ICP, and PbtO_2_ when administered in patients with ABI, i.e. those suffering from subarachnoid hemorrhage (SAH), ICH, traumatic brain injury (TBI), and hypoxic-ischemic encephalopathy (HIE).

## 2. Materials and Methods

### 2.1. Population, Interventions, Comparatives, and Outcomes (PICOs) and Eligibility Criteria

The population (P) of interest were adult (>18 years of age) patients affected by SAH, ICH, TBI, and HIE. The intervention (I) evaluated was the administration of acetazolamide either for diagnostic or therapeutic purposes in the selected population, without comparison (C) with other treatments. The outcomes (O) of interest were changing of CBF and its surrogate (i.e., VMR), ICP and/or PbtO_2_.

According to the PICOs approach, a study was considered eligible if it fulfilled some specific inclusion and exclusion criteria. Clinical trials (either retrospective or prospective) could be enrolled, while case reports, reviews, and letters to editors were excluded. Studies had to be conducted on adult (>18 years of age) humans; studies including children or performed using animals were discarded. The subjects had to be affected by SAH, ICH, TBI, or HIE, whereas other causes of brain injury or healthy subjects were excluded. The administration of acetazolamide was an inclusion criterion regardless of the dosage, the number of doses, the route of administration, and the indications. Furthermore, studies had to report the changes in CBF, ICP, or PbtO_2_ after acetazolamide administrations. There were no restrictions on the method used to measure these variables. Finally, only articles available in English, Italian, or French were considered eligible.

### 2.2. Search Strategies

A comprehensive, computer-based literature research was performed on the PubMed platform. It was structured using the Medical Subject Heading (MeSH), and the keywords which could be contained either in the title or abstract were as follows: “acetazolamide AND brain”, “acetazolamide AND brain oxygenation”, “acetazolamide AND brain tissue oxygenation”, “acetazolamide AND blood flow”, “acetazolamide AND vasomotor reactivity”, “acetazolamide AND intracranial pressure”, “acetazolamide AND intracerebral hemorrhage”, and “acetazolamide AND intracranial hemorrhage”, “acetazolamide AND subarachnoid hemorrhage”, “acetazolamide AND traumatic brain injury”, and “acetazolamide AND hypoxic-ischemic encephalopathy”.

### 2.3. Selection of Studies

All the results obtained from our research were collated in an Excel datasheet and were independently screened according to the PICOs and eligibility criteria by two authors (CS and AH). Any discrepancy was resolved through discussion with the third author (FST). The selection process was completed following the Preferred Reporting Items for Systematic Reviews and Meta-Analyses (PRISMA) recommendations [15] and listed in a flow diagram. Initially, all the duplicates were deleted, and initial selection was performed according to the titles of the articles, discarding the articles that contained at least one exclusion criterion. The next step was selection according to the articles’ abstracts, which consisted of eliminating those that did not report any of the inclusion criteria or had at least one exclusion criterion. The remaining articles were assessed for eligibility via full-text analyses, and a brief justification was reported for exclusion.

### 2.4. Assessment of Risk of Bias of Included Studies

The risk of bias in the included studies was assessed using the Methodological Index For Non-Randomized Studies (MINORS) criteria [16]. Eight or twelve items were evaluated for non-comparative and comparative studies, respectively. Each of them were assigned a score of 0 (unclear), 1 (inadequate), or 2 (adequate).

### 2.5. Data Extraction

Once the inclusion process was completed, the selected articles were analyzed only for a qualitative analysis of the data. All the results were reported and discussed according to the main outcome of the studies. The level of evidence (LOE) was assessed independently by two authors (CS and AH) and reviewed by a third author (FST), according to the Grading of Recommendations, Assessment, Development and Evaluations (GRADE) system [17]. Because of the scarce amount of data and the heterogeneity of the selected studies, no further data synthesis and prioritization of outcome description were reported.

The review was not declared on PROSPERO because of the expected low number of available studies to make a systematic summary of the effects of acetazolamide in this setting.

## 3. Results

### 3.1. Study Selection and Characteristics

From the initial search, 3754 records were identified. After the exclusion of duplicates, 2299 remaining articles were assessed by title and therefore by abstract; among the 25 articles selected for full-text analyses, 14 were further excluded (among those, 3 articles without full text available). The remaining 11 articles were selected for the qualitative analysis. The main results are summarized in Figure 2 and Table 1.

The selected articles were all single-center, non-randomized, prospective interventional studies. The study cohorts ranged between consisting of 15 and 79 subjects, leading to a total of 448 patients. Most of the trials involved subjects with SAH; two studies evaluated patients with ICH [18,19], and one study only focused on patients with HIE [20]. Details of the risk of bias assessment and LOE are reported in Appendix A; all the studies had low LOE.

### 3.2. Summary of Main Results

Acetazolamide was administered intravenously in all studies. The most frequent dose was 1000 mg, given as bolus; different doses were also used, including 500 mg [21], 17 mg/Kg [18], and 1000 mg for body weight < 80 Kgs or 15 mg/kg for body weight > 80 Kgs for a maximum of 1500 mg [22]. In the studies enrolled, no systematic differentiation of the dose according to the pathology analyzed was observed.

**Table 1 brainsci-13-01678-t001:** Summary of the main results of the selected studies.

Author (Year)	Study Design and Population	Outcomes	Measuring Method	Main Results
Shinoda et al. (1991) [23]	Prospective interventional, single-center, Japan 42 patients with SAH	Cerebral perfusion after acetazolamide associated with different findings in the acute and subacute stages of SAH.	^123^I–IMP–SPECT	Diffuse brain swelling: no perfusion in the acute-subacute stages. VMR was reduced in all patients in the acute stage after surgical intervention but improved over time. VMR decreased more in the subacute stage (9–21 days), and it was more frequently reduced in patient with worse clinical status.
Kimura et al. (1993) [24]	Prospective interventional, single-center, Japan 79 patients with SAH	Correlation between VMR and development of DCI due to vasospasms.	^123^I–IMP–SPECT	Reduced VMR by day 8 after SAH in several territories is associated with DCI due to vasospasms.
Yoshida et al. (1996) [25]	Prospective interventional, single-center, Japan 50 patients with SAH	Relationship between VMR in the acute phase and clinical outcome.	Xe-CT	Higher VMR was associated with better clinical outcomes, while there were no differences according to SAH grades or the development of vasospasms.
Kitahara et al. (1996) [18]	Prospective interventional, single-center, Japan 22 patients with ICH	VMR in patients with HPH compared to hypertensive patients with no ICH (non-HPH).	Xe-CT	VMR was preoperatively reduced in the ipsilateral hemisphere compared to non-HPH, while in the thalamus, there were no significant differences. Contralateral VMR in the hemisphere was lower in HPH during the preoperative phase compared to non-HPH, though it significantly rose chronically.
Tanaka et al. (1996) [19]	Prospective interventional, single-center, Japan 15 patients with ICH	VMR in the chronic stage of putaminal vs. thalamic hemorrhages.	Xe-CT	VMR was statistically significant in the chronic stages in both types of hemorrhage.
Szabo et al. (1997) [26]	Prospective interventional, single-center, Hungary 27 patients with aSAH	VMR in a 1–8-year follow-up after a vasospasm in aSAH.	TCD	VMR was restored to normal values.
Tanaka et al. (1998) [27]	Prospective interventional, single-center, Japan 18 patients symptomatic for vasospasms after aSAH and 27 patients asymptomatic for vasospasm after aSAH (tot = 45 patients)	VMR in patients with and without ischemic symptoms due to vasospasms in the acute, subacute, and chronic stages of aSAH.	Xe-CT	In symptomatic patients, VMR was normal in the acute stage, while it was significantly higher in the chronic phase compared to healthy controls (subacute stage was not tested). In asymptomatic patients, VMR was significantly reduced from the acute to the subacute stage, though it rose chronically.
Chang et al. (2003) [21]	Prospective interventional, single-center, Japan 48 patients with ventriculomegaly after aSAH	VMR in a follow-up (1–12 months) compared to healthy subjects and after surgical shunting in patients who developed symptomatic NPH.	Radionuclide angiography with 99mTc-HMPAO	VMR was reduced in asymptomatic and symptomatic patients, except in non-responders to surgery. After shunting, VMR was increased in clinically recovered patients but stable in those who remained symptomatic.
Nogami et al. (2004) [20]	Prospective interventional, single-center, Japan 17 patients with HIE in the subacute stage	Correlation of VMR with clinical outcome; correlation of MRI findings and clinical outcome; correlation of VMR with MRI patterns.	Xe-CT	VMR resulted in more patients with a good clinical outcome. Patients with an unfavorable MRI pattern (hyperintense lesions in T1 and T2) had lower VMR.
Jarus-Dziedzic et al. (2011) [28]	Prospective interventional, single-center, Poland 24 patients with aSAH	BFV and CVR in a long-term follow-up; CO_2_ reactivity in patients treated with clipping, coiling or conservatively.	TCD	VMR, after acetazolamide administration, was restored in the chronic stage. Reactivity to CO_2_ was preserved in the three groups with not statistically significant differences.
Bøthun et al. (2019) [22]	Prospective interventional, single-center, Norway 42 patients with aSAH and 37 patients with UIA (tot = 79 patients)	CVR as a potential predictor of DCI; relationship between CVR and rupture status of the aneurysm (UIA vs. aSAH).	TCD	A reduction in contralateral VMR is predictive of the development of clinical DCI but not radiological infarction. VMR is reduced in patients with aSAH compared to UIA.

123I–IMP–SPECT: N-isopropyl-123iodine single-photon emission computed tomography; 99mTc-HMPAO: 99mTc-hexamethylpropyleneamineoxime; aSAH: aneurysmatic Subarachnoid Hemorrhage; DCI: delayed cerebral ischemia; DFV: Diastolic Flow Velocity; HIE: Hypoxic Ischemic Encephalopathy; HPH: Hypertensive Putaminal Hemorrhage; ICH: Intracranial Hemorrhage; MFV: Mean Flow Velocity; MRI: magnetic resonance imaging; PI: Pulsatility Index; RI: Resistivity Index; SAH: Subarachnoid Hemorrhage; SFV: Systolic Flow Velocity; TCD: transcranial Doppler; VMR: cerebral vasomotor reactivity; Xe-CT: xenon–computed tomography; UIA: Unruptured Intracranial Aneurysm.

All the studies reported data on the effects of acetazolamide on VMR. In the acute stage of SAH (0–4 days), an increase in CBF (measured with Xe-CT) was not different from healthy volunteers, but it was significantly reduced in the subacute (i.e., 5 to 21 days) when compared to the acute phase (increase in CBF 36.1 ± 12.3% vs. 15.4 ± 8.3%, *p* < 0.01, respectively) [27]. Furthermore, a study conducted using N-isopropyl-123iodine single-photon emission computed tomography (123I-IMP SPECT) showed that the worse the clinical presentation, the higher the incidence of reduced VMR, with a peak in the timeframes of day 9–14 and day 15–21 (68% for Hunt–Hess grades I and II; 80–91% for grades III, IV, and V) [23]. When the acute and subacute phase were studied together, VMR in SAH was significantly lower compared to patients with unruptured intracranial aneurism [22]. In the chronic stages of SAH (>21 days), when the response of CBF to acetazolamide was tested via Xe-CT, it was significantly increased compared to the subacute phase (from 15.4 ± 8.3% to 42.3 ± 6.9%, *p* < 0.001) [27], and it was within normal values when ultrasound assessment was performed upon long-term follow-up [26,28].

The subsequent development of vasospasms as a consequence of SAH had no impact on VMR measured in the acute phase using a Xe-CT scan [25], whereas the detection of reduced VMR by day 8 after SAH was associated with distant hypoperfusion and with an augmented risk of developing delayed cerebral ischemia (DCI) in several territories [24]. Moreover, the comparison of VMR of two hemispheres was able to predict the risk of symptomatic DCI (OR 0.96, CI 0.93–1.00, *p* = 0.05 for the lowest value, and *p* = 0.03 for the within-patient average) [22]. Notably, some authors did not test VMR during the subacute phase in patients with symptoms of vasospasms due to “the risk of deteriorating ischemia due to the steal phenomenon” [27]. Finally, it was found that VMR was chronically reduced among patients who subsequently developed chronic hydrocephalus; VMR significantly improved after ventriculoperitoneal shunting in patients clinically improving after surgery, while it remained reduced in those still symptomatic [21]. In patients with putaminal ICH, VMR measured with Xe-CT was significantly reduced in the ipsilateral hemisphere; both hemispheric and thalamic VMR gradually increased in the chronic stages after bleeding [18,19].

Regarding the clinical outcome, VMR was at higher values in patients with good neurological recovery when compared to those with poor clinical evolution, both in SAH (154.7 ± 4.75% vs. 116.9 ± 1.55%, *p* < 0.0001, respectively) [25] and in the subacute phase of HIE (13.3 ± 3.4 vs. 6.8 ± 5.6 mL/100g/min, *p* < 0.05, respectively) [20]. In HIE patients, there was a significant correlation between radiological findings at MRI suggestive of a poor neurological outcome and low VMR values [20].

No data on ICP or PbtO_2_ values were reported in any of the studies.

## 4. Discussion

In this systematic review, we have underlined how limited the data available on the effects of acetazolamide on brain hemodynamics in brain injury patients are. Acetazolamide is a well-known non-competitive inhibitor of carbonic anhydrase, and its effects on various isoenzymes contribute to different organ-specific effects that are not fully understood [29]. Notably, the expression of this enzyme in CNS cells, including neurons, oligodendrocytes, astrocytes, and choroid plexus cells, explains the involvement of acetazolamide in the regulation of vascular tone (and consequently CBF), CSF production, and the development of brain edema, with some effects on aquaporins channels [2]. According to the Monroe–Kellie doctrine [30], all these factors significantly influence ICP. Therefore, assessing the actual impact of acetazolamide in individuals at risk of intracranial hypertension, such as patients with ABI, is crucial. Furthermore, given the emerging understanding of the relationship between ICP and PbtO_2_ [31,32,33], further investigations into the potential effects of acetazolamide on brain oxygenation in this context are warranted.

We found that there were no studies that measured the impact of acetazolamide on ICP. Despite its administration in the ABI population, there is currently no evidence supporting a significant change in ICP, its duration, or any clinical implications. This information is noteworthy because a study in healthy animal models demonstrated that after a transient increase, acetazolamide led to a reduction in ICP [3]. It was further demonstrated that this effect was directly attributed to the reduction in CSF production rather than other potential mechanisms (such as compensatory changes in minute ventilation to restore baseline pH or fluid excretion by the kidneys). These results were obtained in anesthetized, ventilated, and nephrectomized rats [3]. A similar trend towards a reduction in ICP was observed in another model using healthy rats, although it did not reach statistical significance [34]. Moreover, when studied in a rat model of ICH, acetazolamide demonstrated a protective effect by reducing ICP spikes through decreased CSF production, resulting in increased cerebral compliance [4]. While a reduction in CSF production induced by acetazolamide is well understood in humans and has shown success in the treatment of idiopathic intracranial hypertension and CSF leaks [1], evidence regarding its use in ABI patients is still lacking.

All the findings from our systematic review focused on another key determinant of ICP, namely CBF. The effect of acetazolamide on CBF was demonstrated in the 1960s and 1970s, when Severinghaus et al. [6] and Laux et al. [8] showed an increase in CBF following acetazolamide administration in dogs and rhesus monkeys, respectively. In subsequent years, the evaluation of CBF reactivity to acetazolamide was utilized as a prognostic tool in patients with cerebrovascular diseases (such as carotid artery occlusion or ischemic stroke) since a reduced VMR was associated with an increased risk of ischemic events in the affected regions [7]. Similarly, an altered vasodilation response to acetazolamide was also observed in patients with ABI. Interestingly, the timing of VMR variation induced by acetazolamide followed a specific pattern in the course of ABI, particularly after SAH. Reports on vasodilation induced by acetazolamide in the acute phase have conflicting results, with some trials showing a reduced response [22,23] and others demonstrating a preserved response [27]. However, during the subacute stage (i.e., 5–21 days), consistently reduced VMR was observed [22,23]. This phase occurs after the development of the so-called “early brain injury” (EBI), which encompasses the pathophysiological events occurring within the first 72 h after SAH, including increased ICP, cerebral hypoperfusion, neuroinflammation, and spread depolarizations [35]. EBI can lead to secondary brain injury, and it lays the foundation for the development of vasospasms and delayed cerebral ischemia (DCI) [35]. Therefore, it is plausible to assume that the reduction in VMR in response to acetazolamide is one manifestation of EBI onset. This hypothesis is supported by the fact that a more pronounced decrease in VMR is associated with patients with poor clinical outcomes [25] and developing DCI [22,24].

Various possible explanations have been proposed for the reduction in vasodilatory response during early brain injury after SAH; some authors have suggested that the ischemia and hypoxia occurring in the affected regions trigger a metabolic cascade, leading to the release of vasoactive substances that induce compensatory vasodilation in order to adequately supply the hypoperfused cerebral parenchyma. Consequently, this could limit the reactivity to another vasodilatory stimulus [27]. However, recent studies have highlighted the significance of vasoconstriction in relation to the vasodilatory reserve [36]. For instance, in a rat model, Balbi et al. demonstrated a paradoxical vasoconstriction of pial and parenchymal vasculature in response to CO_2_ inhalation during the first 24 h after SAH induction [37]. This altered response may partially account for the impairment of VMR following acetazolamide administration. Another possible mechanism is the direct impact of increased pressure on the vessel walls caused by factors such as blood [18], CSF [21], cerebral edema [38], and the resultant elevated ICP. This hypothesis is supported by some of the findings of this review. Specifically, ICH, CBF, and VMR in the thalamus were reduced in patients with these conditions in the preoperative stage and increased following decompression with surgical hematoma evacuation; although only the increase in CBF reached statistical significance, the VMR in response to acetazolamide followed the same trend [18]. Furthermore, extrinsic compression exerted by CSF has also been postulated as a potential contributing factor. In patients with normal-pressure hydrocephalus after SAH, successful surgical shunting leading to clinical resolution resulted in the restoration of previously reduced VMR [21].

Understanding the variation in ICP and its determinants after acetazolamide administration is important due to the relationship between PbtO_2_ and cerebral perfusion pressure, which is another main determinant of cerebral oxygen delivery. Manipulating CPP by increasing MAP or reducing ICP can improve regional CBF in ischemic areas [39]. Although our review did not identify any trials describing the effect of acetazolamide on PbtO2 in ABI, insights can be extrapolated from studies conducted on animals [40,41], patients without ABI [10,12], and healthy volunteers [42] demonstrating a desirable increase in brain oxygenation. Bickler et al. [41] and Kealy et al. [40] provided evidence that the increase in PbtO_2_ induced by acetazolamide was primarily due to the elevation of CBF and thus cerebral DO_2_ rather than other mechanisms (i.e., the Bohr effect on the hemoglobin dissociation curve). Moreover, a study utilizing near-infrared spectroscopy in healthy volunteers and patients with acute ischemic stroke showed that acetazolamide increased regional oxygen saturation, which was directly correlated with regional CBF [12]. Additionally, following acetazolamide administration, an increase in cerebral oxygen delivery was not accompanied by a proportionally elevated metabolic demand, leading to a reduction in the oxygen extraction fraction. This finding was reported by Vorstrup et al. [10] through measurements of jugular vein oxygen saturation and by Buch et al. [42] in a study involving five healthy volunteers undergoing magnetic resonance imaging. These studies suggested a constant cerebral metabolic rate of oxygen, indicating that the increase in DO2 was directly influenced by acetazolamide’s effects on CBF rather than an elevated metabolic demand.

This systematic review has several limitations. Firstly, the scarcity of available studies and the diversity among the included studies (such as variations in CBF measurement methods, disparate outcomes assessed, and differences in disease stages) rendered it infeasible to conduct a qualitative analysis of the measured outcomes. Secondly, our research was restricted to a single database. However, despite these aforementioned limitations, the main strength of this review lies in its ability to underscore the necessity for additional studies concerning the impact of acetazolamide on ABI patients. Indeed, while some studies, encompassing both animal and non-animal research, have indicated a potential beneficial effect of acetazolamide administration on CBF, ICP, and PbtO_2_, there exists a dearth of high-quality investigations providing dependable data on the variation in these three variables in ABI. Specifically, further research holds the potential to enhance our comprehension of the mechanisms underpinning cerebral vasoreactivity in ABI, elucidate the potential adverse effects associated with the drug, and subsequently safely harness the potential therapeutic benefits of acetazolamide. Future research should direct its attention towards several key aspects, including (a) investigating whether the effects of acetazolamide are dose-dependent, (b) assessing the potential variations in these effects among different forms of brain injury, (c) examining the impact of acetazolamide on tissue oxygenation and metabolism, (d) determining the duration of these effects, and (e) identifying and evaluating potential side effects. Conducting observational physiological studies in these areas may offer a more comprehensive understanding of how acetazolamide therapy could be effectively integrated into the management of patients with brain injuries.

## 5. Conclusions

In this systematic review, we identified limited data on the effects of acetazolamide on cerebral hemodynamics after an acute brain injury. In particular, no data on ICP and PbtO_2_ are available, emphasizing the need for high-quality physiological studies in this context.

## Figures and Tables

**Figure 1 brainsci-13-01678-f001:**
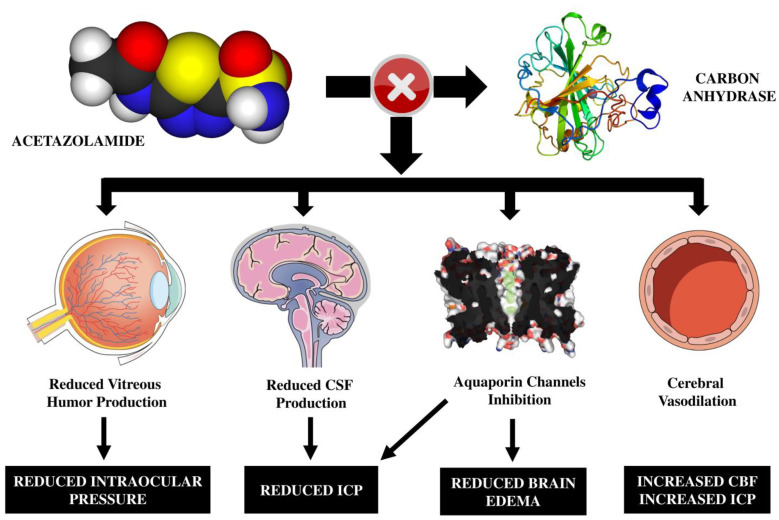
A schematic representation of the effects of acetazolamide on brain function. ICP = intracranial hypertension; CSF = cerebrospinal fluid.

**Figure 2 brainsci-13-01678-f002:**
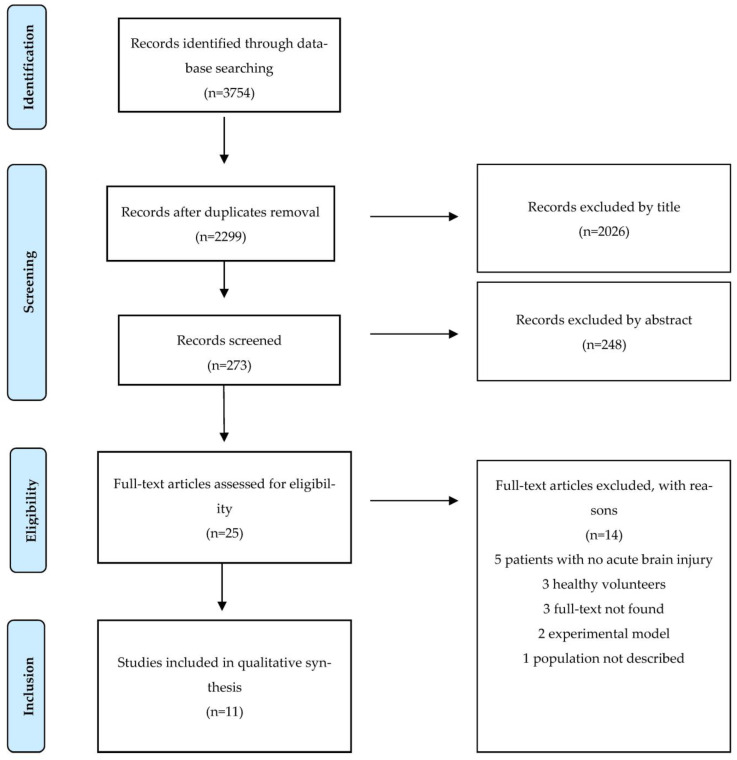
Flowchart of the selection of the studies analyzed.

## Data Availability

Data are available from the corresponding author upon request. The data are not publicly available due to privacy and ethical restrictions.

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
