# Peer review of "The Effects of Acetazolamide on Cerebral Hemodynamics in Adult Patients with an Acute Brain Injury: A Systematic Review"

_brainsci, 2023, doi:10.3390/brainsci13121678_

Round 1

Reviewer 1 Report

Comments and Suggestions for Authors

In this review the authors are systematically reviewing literature on the impact of Acetazolamide on cerebral blood flow and reactivity in acute brain injury conditions. This is an interesting review that highlights the role of Acetazolamide in brain oxygenation and perfusion after acute brain injury. This would be of interest of readers of brain sciences. The following concerns needs to be addressed before the publication.

Major Concern

1-     Although the nature of this review is rather qualitative than quantitative, statistical analysis would strengthen the quality of the report. This could include test for normality of data, summary intervention estimate, and other statistical analysis that are being recommended for Meta analysis.

Minor Concerns

2-     Search keywords could be started with broader terms such as “Acetazolamide and CNS”, “Acetazolamide and brain”. Also inclusion of brand name of Acetazolamide may bring some resources.

3-     Selection of studies. It would be helpful to list the inclusion exclusion criteria if existed. This would help to clarify unbiased selection of articles.

4-     It is of interest if the data on the dose could be classified based on the pathology.

Author Response

  1. In this review the authors are systematically reviewing literature on the impact of Acetazolamide on cerebral blood flow and reactivity in acute brain injury conditions. This is an interesting review that highlights the role of Acetazolamide in brain oxygenation and perfusion after acute brain injury. This would be of interest of readers of brain sciences. The following concerns needs to be addressed before the publication.

Authors’ response = Thanks for your nice comment.

  1. Although the nature of this review is rather qualitative than quantitative, statistical analysis would strengthen the quality of the report. This could include test for normality of data, summary intervention estimate, and other statistical analysis that are being recommended for Meta analysis.

Authors’ response = We thank the reviewer for this proposal. However, as indicated into the qualitative description of the existing data, this is not possible as data on cerebrovascular reactivity (the solely available) are very heterogenous and combining them together would not be feasible.

  1. Search keywords could be started with broader terms such as “Acetazolamide and CNS”, “Acetazolamide and brain”. Also inclusion of brand name of Acetazolamide may bring some resources.

Authors’ response = In order to intercept as many articles as possible, we did search the keywords in both title and abstracts, starting with “acetazolamide and brain”. The research with “Acetazolamide and CNS” brought even less articles and none which was not identified by the initial search. For what concerns the brand name, it is an important cue as the name Diamox (the only approved) is very popular in the scientific community. By the way, the research of “Diamox NOT acetazolamide”, without any other restriction, brought out only 286 results and none of these reflected our PICO’s. For this reason, the research was not extended. Anyway, we added it as a limitation of the study.

  1. Selection of studies. It would be helpful to list the inclusion exclusion criteria if existed. This would help to clarify unbiased selection of articles.

Authors’ response = We tried to clarify the inclusion/exclusion criteria in the lines 95-105

  1. It is of interest if the data on the dose could be classified based on the pathology.

Authors’ response = Unfortunately, there was not any systematic differentiation of dose according to the pathology analyzed. We did add this datum on the paper (lines 188-189).

Reviewer 2 Report

Comments and Suggestions for Authors

Dear Authors,

I have read your review with interest. Please find some suggestions/advces below:

1. There is some mess about Authors and Affiliations ( there are 3 numbers of affiliations and all the Authors are affiliated ad No. 1) It is to be clarified

2. I would enrich introduction with slightly more detailed description of VMR measurements with TCD or Xe-CT

3. As most of cases in analyzed papers are subjects with SAH, could you please provide some data on relation between therapy with acetazolamide and incidence of vasospasm?

Comments on the Quality of English Language

Minor English editing is suggested.

Author Response

  1. Dear Authors, I have read your review with interest. Please find some suggestions/advices below. There is some mess about Authors and Affiliations ( there are 3 numbers of affiliations and all the Authors are affiliated ad No. 1) It is to be clarified.

Authors’ response = Thank you for your comments and nice cues.

We made up the affiliation problem, as Dr Stella is affiliated to nr 1 and 2, while Dr Schuind is affiliated to nr 3.

  1. I would enrich introduction with slightly more detailed description of VMR measurements with TCD or Xe-CT

Authors’ response = We did agree on the importance of these two techniques; thus, we added lines 59-72.

  1. As most of cases in analyzed papers are subjects with SAH, could you please provide some data on relation between therapy with acetazolamide and incidence of vasospasm?

Authors’ response = We do agree with the importance of highlighting any possible correlation between acetazolamide administration and vasospasm. Unfortunately, we did not find any study regarding either short- or long-term therapy with acetazolamide. In fact, the articles that met the including criteria were based on the administration of one single dose of acetazolamide used to analyze the variation of CBF. By the way, what we were able to find is reported in lines 182-189 (we added one information about stealing phenomenon) and discussed in lines 248-257.

  1. Minor English editing is suggested.

Authors’ response = We have revised the manuscript, accordingly.

Reviewer 3 Report

Comments and Suggestions for Authors

Authors have described “The effects of acetazolamide on cerebral hemodynamics in adult patients with an acute brain injury: A systematic review”.  The concept of MS is good. However, the Introduction section can be improved incorporating more knowledge related to the field.

The specific comments, which could help to improve the manuscript, are:

1.      The manuscript should be revised for grammatical & punctuation errors.

2.      Authors have used PUBMED search. Why authors did not use another databases like EMBASE (via Ovid), and the Cochrane library.

3.      I would appreciate to make following aspects clear: which articles were only screened for headings, which were screened for the abstract and which were finally read in full. This would mean a significant methodological upgrading of the publication. This upgrading would also suggest the formulation of clear inclusion and exclusion criteria, which I hope will have been established before work begins.

4.      The result section is very short. It would be better to add more related to the title.

5.      Authors are suggested to add limitation of study.

6.      It would be better to add a figure depicting acetazolamide mechanism of action on cerebral hemodynamics/acute brain injury.

7.      It would be better to add research gap and future prospects related to the topic in conclusion section.

Comments on the Quality of English Language

Moderate editing of English language is required.

Author Response

  1. Authors have described “The effects of acetazolamide on cerebral hemodynamics in adult patients with an acute brain injury: A systematic review”.  The concept of MS is good. However, the Introduction section can be improved incorporating more knowledge related to the field.

Authors’ response = Thank you for your comments and nice cues. We did enrich our introduction

  1. The manuscript should be revised for grammatical & punctuation errors.

Authors’ response = We have revised the manuscript, accordingly.

  1. Authors have used PUBMED search. Why authors did not use another databases like EMBASE (via Ovid), and the Cochrane library.

Authors’ response = This is a strong limitation of the design of the study. We added it in the discussion section. However, no other articles was identified also in published reviews on the topic. Considering the scarcity of existing data, we do not think that the quality or novelty of findings would be improved by searching other databases.

  1. I would appreciate to make following aspects clear: which articles were only screened for headings, which were screened for the abstract and which were finally read in full. This would mean a significant methodological upgrading of the publication. This upgrading would also suggest the formulation of clear inclusion and exclusion criteria, which I hope will have been established before work begins.

Authors’ response = In order to clarify the screening and selection process, we listed out the inclusion and exclusion criteria (lines95-105). Furthermore, the progressive approach to the enrollment is now more detailed at lines 129-132, while the results are reported in figure 1.

  1. The result section is very short. It would be better to add more related to the title.

Authors’ response = We thank the reviewer for this comment. However, the available data are scarce and no other additional issues related to the research outcomes were available. We do believe there is a lot of investigations to develop on this topic.

  1. Authors are suggested to add limitation of study.

Authors’ response = Limitations of the study had been added in the discussion section.

  1. It would be better to add a figure depicting acetazolamide mechanism of action on cerebral hemodynamics/acute brain injury.

Authors’ response = We have developed a Figure (Figure 1), accordingly.

  1. It would be better to add research gap and future prospects related to the topic in conclusion section.

Authors’ response = As you underlined, we do believe there are some prospects related to the topic and we agree it was important to add them. By the way, we expressed what the research gaps and possible future research areas are in the discussion section, because we believe that these gaps are among the most important forces in the review.

Round 2

Reviewer 3 Report

Comments and Suggestions for Authors

The authors have justified the comments. However, it would be better to add a crisp of the research gap and future prospects in the conclusion section.

Author Response

The authors have justified the comments. However, it would be better to add a crisp of the research gap and future prospects in the conclusion section.

Authors' response: This has been added, as requested.